# Comparison of Three Catalytic Processes in Degradation of HPAM by tBu-TPyzPzCo

**Dejun Wang** [1,2,3], **Hui Li** [4], **Xiren Jiang** [1,2,*], **Chaocheng Zhao** [3] and **Yuhui Zhao** [1,2]

1    North China Sea Environmental Monitoring Center, State Oceanic Administration, Qingdao 266033, China; wdjhello@outlook.com (D.W.); zhaoyuhui@ncs.mnr.gov.cn (Y.Z.)
2    Shandong Provincial Key Laboratory of Marine Ecology and Environment & Disaster Prevention and Mitigation, Qingdao 266033, China
3    College of Chemical Engineering, China University of Petroleum, Qingdao 266580, China; zhaochch@upc.edu.cn
4    Human Resource Department, Qingdao Technical College, Qingdao 266555, China; lihui@qtc.edu.cn
*    Correspondence: jiangxiren@ncs.mnr.gov.cn; Tel.: +86-0532-58761082

**Abstract:** The present study describes a two-step synthesis process for the cobalt complex of tetra-2,3-(5,6-di-tert-butyl-pyrazino) porphyrazine (*t*Bu-TPyzPzCo). The product was ultrasonically impregnated onto carbon black (CB) to prepare a supported catalyst (*t*Bu-TPyzPzCo/CB). We built a split photoelectric catalytic device to test the performance of photocatalytic, electrocatalytic and photoelectrocatalytic degradation of partially hydrolyzed polyacrylamide (HPAM). The results confirm that HPAM exhibited more efficient degradation in the presence of a supporting catalyst using the photoelectrocatalytic process than by photocatalytic or electrocatalytic oxidation—or even the sum of the two in saline water. The photoelectrocatalytic reaction confirmed that the process conforms to quasi-first order reaction kinetics, while the reaction rate constants were 6.03 times that of photocatalysis and 3.97 times that of electrocatalysis. We also compared the energy consumption of the three processes and found that the photoelectrocatalytic process has the highest energy efficiency.

**Keywords:** cobalt azaphthalocyanine; catalytic processes; degradation; polyacrylamide; comparison



## 1. Introduction

The sewage treatment technologies most commonly used in current production and life practices mainly include physical, chemical, and biological methods. Among them, biological methods are the primary technology, and the other two are supplementary. In practical applications, the three have some shortcomings. The application of photoelectrocatalytic oxidation technology is less limited, and it is more suitable for the treatment of refractory organic wastewater [1–5]. The actual operation of photoelectrocatalytic oxidation technology can be subdivided into photocatalytic oxidation technology, electrocatalytic oxidation technology and photoelectrocatalytic oxidation technology [6–8].

Research on photocatalysis and electrocatalysis technology is quite mature. In recent years, researchers have found that when light-activated semiconductor particles are immersed in an electrolyte containing redox potential components, the electric field of the Schottky barrier formed can make photoelectrons and holes move in opposite directions by means of electron migration, thereby achieving separation of the two [9,10]. Based on this, researchers began to use electrochemical methods to understand the surface characteristics and reaction mechanisms of semiconductors in the photocatalytic degradation process. Within a few years, this research developed into an electrochemically assisted photocatalytic method, called the photoelectrocatalytic method [10,11].

Most previous studies looking at what happens in a specific catalytic process have used new catalysts, so they have mainly focused on the preparation of catalysts and neglected the study of the catalytic process. In this paper, a split photoelectrocatalytic

reactor was designed, using *tert*-butyl substituted cobalt octaazaphthalocyanine supported by conductive carbon black (*t*Bu-TPyzPzsCo/CB) as the catalyst, to study the degradation effects of hydrolytic polyacrylamide (HPAM) in different catalytic processes, and to analyze the reaction processes and mechanisms.

## 2. Results and Discussions

### 2.1. Catalyst Characterization

The UV-visible absorption spectra for *t*Bu-TPyzPzCo and unsubstituted TPyzPzCo are shown in Figure 1, wherein *N*,*N*-Dimethyl Formamide (DMF) was employed as the solvent within the range of 300–800 nm. The compounds exhibited two evident absorption bands at 330 nm, 308 nm, 615.5 nm and 633.5 nm, respectively, which corresponded to the B and Q bands of the phthalocyanine azaanalogues [12,13]. A comparison of the two spectra indicated the presence of *t*Bu-TPyzPzCo absorption peaks, which are represented by a red shift in the UV region and a blue shift in the visible region, possibly due to the larger $\pi$ structure. In addition, several weak peaks near the Q band in the spectrum of the unsubstituted TPyzPzCo that were more intense than *t*Bu-TPyzPzCo were observed. The peaks were attributed to the polymerization-produced dimer [14] and affected the compounds' properties, thereby allowing the presence of peripheral substituents to suppress dimer generation.

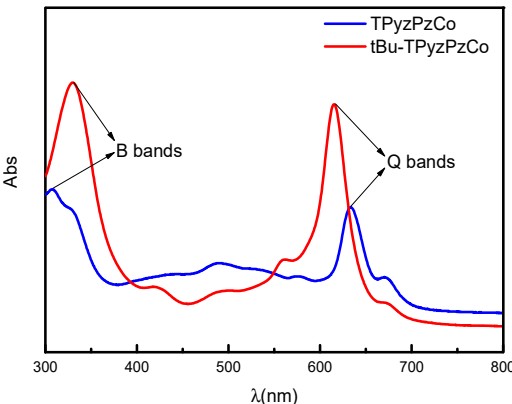

**Figure 1.** UV-visible absorption spectra of unsubstituted TPyzPzCo (blue) and *t*Bu-TPyzPzCo (red) in DMF solution.

The FTIR spectra of *t*Bu-TPyzPzCo before and after loading are presented in Figure 2. They were recorded for the fundamental region of 400–4000 cm$^{-1}$ and used the KBr disk technique. The two spectra exhibited particularly similar absorption peaks around 769, 964, 1122 and 1329 cm$^{-1}$, which may have presented the phthalocyanine skeletal vibrations [15]. Due to the presence of the *tert*-butyls, the spectra also exhibited absorption peaks in the range around 1375 cm$^{-1}$. Specifically, two peaks were observed at 1371 cm$^{-1}$ and 1386 cm$^{-1}$, and the strength of the former was about two-fold that of the latter. In addition, two absorption peaks were observed around 1456 cm$^{-1}$ and 1682 cm$^{-1}$, which represented the C=C and C=N bonds stretching vibration of the pyrazine macrocycles.

Figure 3 displays the morphology of *t*Bu-TPyzPzCo before and after loading, and the energy dispersive X-ray analysis (EDAX) of *t*Bu-TPyzPzCo/CB. The SEM image exhibited random pore size distributions and interconnected pore systems in the compound, wherein a sheet structure with a thickness of about 1 μm was observed. After loading, the loose holes in the carbon black were covered to some extent by the catalyst. In addition, the catalyst was well loaded on the conductive carrier, and the presence of the cobalt element on the EDAX spectrum was observed.

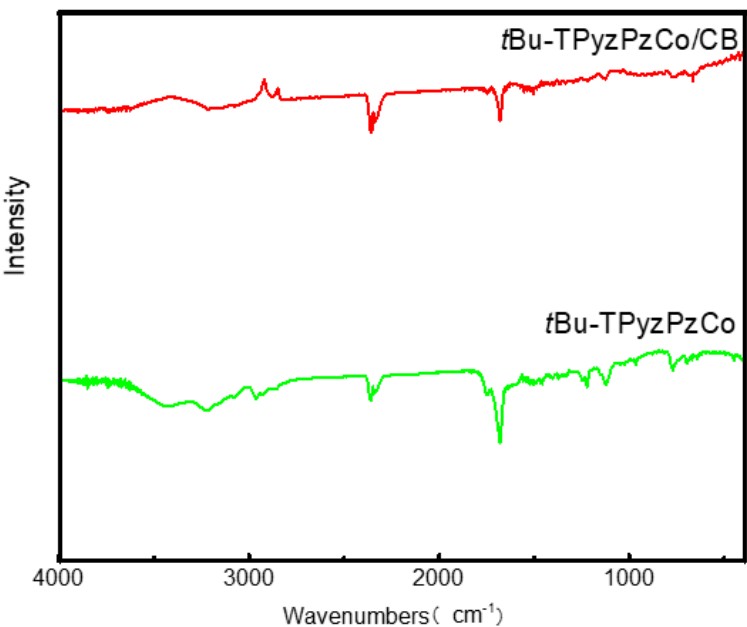

**Figure 2.** FTIR spectroscopy results of *t*Bu-TPyzPzCo before and after loading.

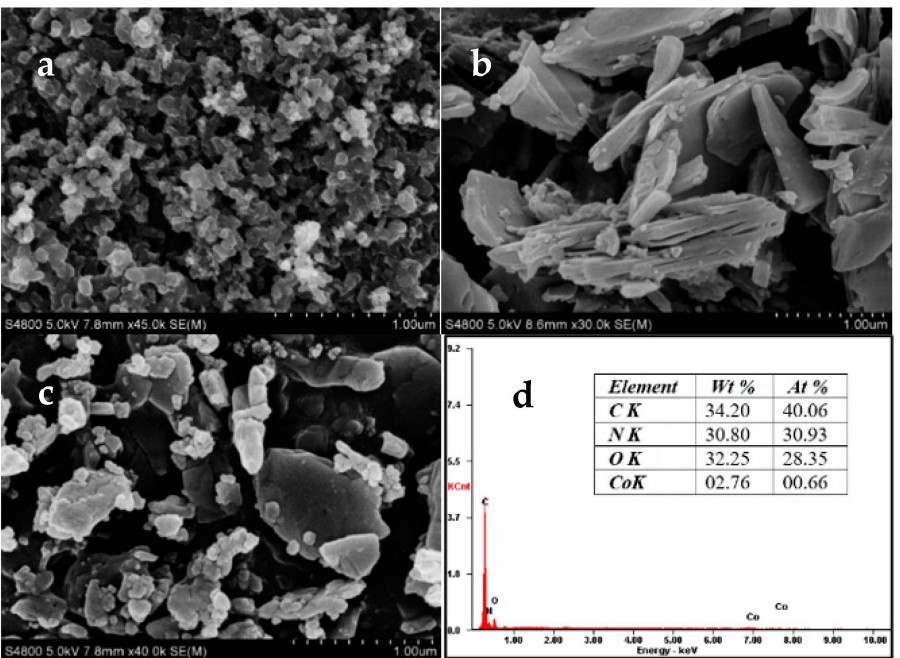

**Figure 3.** SEM images of *t*Bu-TPyzPzCo before and after loading and the energy dispersive X-ray analysis (EDAX) of the supported catalyst. (**a**) Carbon Black; (**b**) *t*Bu-TPyzPzCo; (**c**) *t*Bu-TpyzPzCo/CB; (**d**) *t*Bu-TpyzPzCo/CB.

### 2.2. Photoelectrocatalytic Performance

Figure 4 shows the catalytic performance results of *t*Bu-TPyzPzCo and *t*Bu-TPyzPzCo/CB in different processes. The HPAM removal efficiency and viscosity of the unsupported catalyst of each catalytic method were significantly lower than those of the supported catalyst. For *t*Bu-TPyzPzCo/CB, the degradation efficiency of HPAM was less than 50% after 2 h of degradation in a single catalytic process. In contrast, the ratio of photoelectrocatalysis was 94.55%, which was much higher than the sum of the other two. This fact indicated that a synergistic effect existed in the combination process, along with several factors which influenced HPAM viscosity (such as temperature, concentration, and molecular weight). In

this study, the initial viscosity of the 50 mg/L HPAM solution was 8.33 mPas, which decreased to 5.62 mPas because of the addition of the electrolyte and catalyst after magnetic stirring for half an hour. The viscosity of the solution was easily affected by anions and stirring. As the reaction progressed, HPAM gradually degraded, and the viscosity of the solution decreased accordingly. This downward trend was consistent with the degradation efficiency.

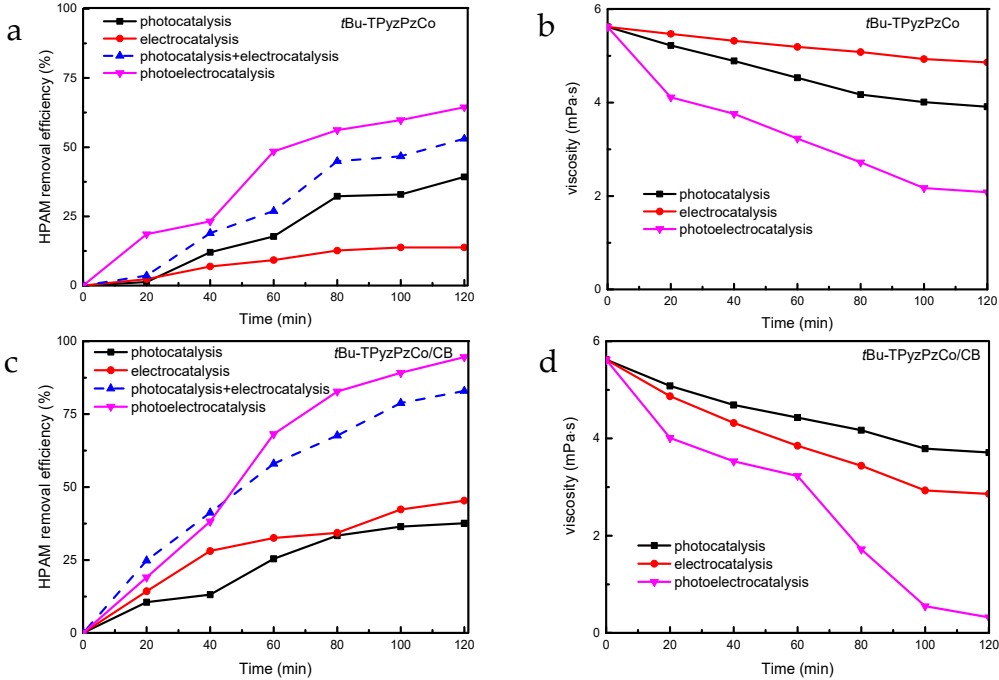

**Figure 4.** Comparison of degradation of hydrolyzed polyacrylamide (HPAM) by the different catalytic processes. (**a**) *t*Bu-TpyzPzCo; (**b**) *t*Bu-TpyzPzCo; (**c**) *t*Bu-TpyzPzCo/CB; (**d**) *t*Bu-TpyzPzCo/CB. (electrolyte: 0.1 mol/L; initial concentration of HPAM: 50 mg/L; voltage: 40 V; catalyst: 0.3 g/L).

Previous studies have proven that the reaction rate of the photoelectrocatalytic process is affected by many reaction conditions, such as light source and light intensity, bipolar voltage, pollutant concentration, nature of the reaction medium, pH, temperature, pressure and catalyst properties [16,17]. We previously proved that the photoelectrocatalytic processes constructed in this paper can fully perform their respective functions in single photocatalytic and single electrocatalytic processes. They can also use chemical collateral effects between each other to produce synergistic effects and degrade HPAM using the combined force of the oxidation abilities in the system.

The degradation of HPAM in the photoelectrocatalytic process can be divided into paths as shown in Equations (1)–(4), where products 1–4 are just to distinguish between the different reaction processes. It may be the same product in the actual reaction.

$$\text{HPAM} \xrightarrow{k_E} \text{Product1} + CO_2 + H_2O \text{ (Degradation caused by electrocatalysis)} \quad (1)$$

$$\text{HPAM} \xrightarrow{k_P} \text{Product2} + CO_2 + H_2O \text{ (Degradation caused by photocatalysis)} \quad (2)$$

$$\text{HPAM} \xrightarrow{k_{X \to E}} \text{Product3} + CO_2 + H_2O$$
$$\text{(Increased efficiency produced by electrocatalysis affected by xenon lamp)} \quad (3)$$

$$\text{HPAM} \xrightarrow{k_{V \to P}} \text{Product4} + CO_2 + H_2O$$
$$\text{(Increased efficiency produced by photocatalysis affected by voltage)} \quad (4)$$

According to the literature, photocatalytic oxidation [18] and electrochemical degradation of organic pollutants basically follow quasi-first order reaction kinetics. Corre-

spondingly, the increase of oxidized species produced under the photoelectrocatalytic effect also undergoes a homogeneous reaction in the reaction system, so the total rate of photoelectrocatalytic degradation of HPAM in this paper can be expressed as Formula (5):

$$-\frac{\mathrm{d}c}{\mathrm{d}t} = k_{\mathrm{E}} \cdot c + k_{\mathrm{P}} \cdot c + (k_{\mathrm{X-E}} + k_{\mathrm{V-P}}) \cdot c \tag{5}$$

In the equation, the first term on the right side of the equals sign represents the oxidation in the electrocatalytic process, and $k_{\mathrm{E}}$ is the reaction rate of the process; the second term represents the oxidation in the photocatalytic process, and $k_{\mathrm{P}}$ is the reaction rate of the process; the third item represents the synergy produced by the interaction between light and electrical systems in the photoelectrocatalytic process. This third item includes electrocatalytic degradation under xenon lamp irradiation (reaction rate expressed as $k_{\mathrm{X-E}}$), and photocatalytic degradation under applied voltage (reaction rate expressed as $k_{\mathrm{V-P}}$).

Integrate Equation (5) to get (6)

$$c = c_0 \mathrm{e}^{-(k_{\mathrm{E}} + k_{\mathrm{P}} + k_{\mathrm{X-E}} + k_{\mathrm{V-P}})t} \tag{6}$$

In the formula, $c_0$ represents the initial concentration of HPAM; $t$ represents the photoelectrocatalytic degradation time.

Suppose $K = k_{\mathrm{E}} + k_{\mathrm{P}} + k_{\mathrm{X-E}} + k_{\mathrm{V-P}}$, then (6) is simplified to (7):

$$c = c_0 \mathrm{e}^{-Kt} \tag{7}$$

It can be seen from Equation (7) that the photoelectrocatalytic degradation of HPAM in this study is the same as the photocatalytic and electrocatalytic degradation of HPAM. It should also follow quasi-first order reaction kinetics, and the total degradation rate of the reaction is expressed as $K$.

The Langmuir–Hinshelwood reaction formula was employed to linearly fit the results of the three catalytic processes, as presented in Figure 5.

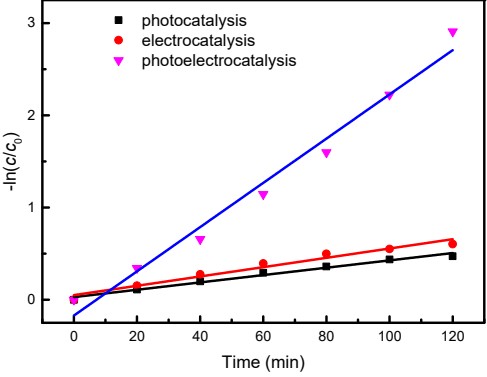

**Figure 5.** Kinetic analysis of the three catalytic processes. (electrolyte: 0.1 mol/L; initial concentration of HPAM: 50 mg/L; voltage: 40 V; catalyst: 0.3 g/L)

The results confirmed that all three HPAM catalytic degradation processes were quasi-first order reactions. The reaction rate constant of the photoelectrocatalytic system was $2.4 \times 10^{-2}$ min$^{-1}$, which was 6.03 times that of photocatalysis and 3.97 times that of electrocatalysis. The reaction rate constants and fitting correlation coefficients of the three catalytic methods are shown in Table 1.

**Table 1.** Kinetics reaction rate constants of the three catalytic processes.

| No. | Cell Voltage (V) | Na$_2$SO$_4$ (mol/L) | Reaction Rate (min$^{-1}$) | $R^2$ | Process |
|-----|-----|-----|-----|-----|-----|
| 1 | - | 0.1 | $3.98 \times 10^{-3}$ | 0.9789 | Photocatalytic |
| 2 | 40 | 0.1 | $5.05 \times 10^{-3}$ | 0.9614 | Electrocatalytic |
| 3 | 40 | 0.1 | $2.4 \times 10^{-2}$ | 0.9771 | Photoelectrocatalytic |

*2.3. Synergy Mechanism Analysis*

In photocatalytic systems, the existing electrochemical indirect oxidation mechanism of phthalocyanine complexes was proposed by Comninellis et al. [19,20] and has been improved on and verified by many scholars [21–23]. In the electrochemical degradation process, pollutants (represented by W) can be directly oxidized to produce oxidizing pollutant cations (W•$^+$), and water molecules on the electrode surface can also be decomposed, forming hydroxyl radicals (•OH) adsorbed on the electrode surface, as shown in Formulas (8) and (9). During the photocatalytic degradation process, the phthalocyanine (represented by P) absorbs sunlight to reach the first singlet excited state $^1$P*, which can undergo intersystem crossing (isc) to the triplet excited state $^3$P*. Both $^1$P* and $^3$P* can react with water molecules to produce •OH, as shown in Formulas (10) and (11) [24–26]. Additionally, both can react with pollutants (W) [27] to give the oxidized form of the pollutant (W•$^+$) and the reduced form of the photocatalyst (P•$^-$), as shown in Formulas (12) and (13). In the electrocatalytic reaction process, the oxygen molecules near the stainless steel cathode gain electrons to generate superoxide radicals O$_2$•$^-$ [28], as shown in Formula (14), and P•$^-$ can also react with molecular oxygen to generate O$_2$•$^-$ [29], as shown in Formula (15). Hydroxyl radicals, superoxide radicals, and oxidizing pollutant cations can all directly oxidize pollutants until mineralization, as shown in Equations (16) to (18).

$$W \rightarrow W\bullet^+ + e^- \tag{8}$$

$$H_2O \rightarrow \bullet OH + H^+ + e^- \tag{9}$$

$$P \overset{h\nu}{\leftrightarrow} {}^1P^* + H_2O \rightarrow P\bullet^- + \bullet OH + H^+ + e^- \tag{10}$$

$$P \overset{h\nu}{\leftrightarrow} {}^1P^* \overset{isc}{\rightarrow} {}^3P^* + H_2O \rightarrow P\bullet^- + \bullet OH + H^+ + e^- \tag{11}$$

$$P \overset{h\nu}{\leftrightarrow} {}^1P^* + W \rightarrow P\bullet^- + W\bullet^+ \tag{12}$$

$$P \overset{h\nu}{\leftrightarrow} {}^1P^* \overset{isc}{\rightarrow} {}^3P^* + W \rightarrow P\bullet^- + W\bullet^+ \tag{13}$$

$$O_2 + e^- \rightarrow O_2\bullet^- \tag{14}$$

$$P\bullet^- + O_2 \rightarrow P + O_2\bullet^- \tag{15}$$

$$W + \bullet OH \rightarrow \text{Oxidation products} \tag{16}$$

$$W + O_2\bullet^- \rightarrow \text{Oxidation products} \tag{17}$$

$$W\bullet^+ + O_2\bullet^- \rightarrow \text{Oxidation products} \tag{18}$$

Through literature review, there may be a degradation pathway, in which the chain scission reaction occurs first, and HPAM is decomposed into various units with relatively low molecular weights [30–32]. According to our research, small molecules were degraded on *t*Bu-TPyzPzCo/CB under xenon lamp irradiation and an external electric field. To study the degradation pathway clearly, we used a multi-instrument combined method to detect intermediates, especially PyGCMS, $^1$HNMR, HSGCMS and EDX. It is worth noting that we did not use HPLC/MS, because high concentrations of electrolyte will cause serious distortion of the results. The results suggested that acrylamide monomer and acrylic acid were intermediate products. A possible path for HPAM photoelectrocatalytic degradation is proposed in Figure 6 based on those characterizations.

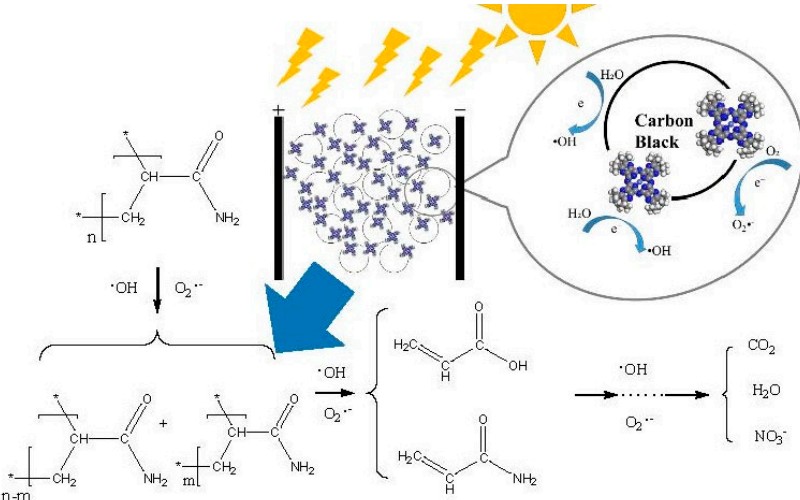

**Figure 6.** Schematic diagram of HPAM degradation reaction by photoelectrocatalysis using *t*Bu-TPyzPzCo/CB particle electrodes.

### 2.4. Energy Consumption Analysis

In actual water treatment applications, while pursuing a higher pollutant removal rate, the operating cost of the system is an important indicator for evaluating the quality of the process. In this study, when photocatalysis, electrocatalysis, and photoelectrocatalysis processes treat HPAM-containing wastewater, energy consumption is the most important operating cost. To compare that of the three processes, this study calculated the running time and energy consumption of the system when the same mass (40%) of HPAM was removed, and obtained energy efficiency. The reaction conditions were as follows: the catalyst addition was 0.3 g/L, the applied voltage across the electrode was 40 v, and the $Na_2SO_4$ electrolyte concentration was 0.1 mol/L. The power of the xenon lamp used in this study was 300 W, higher than the 200 W of the DC power source. The calculation of energy efficiency and energy consumption used the following, Formulas (19) and (20), and the experimental results are shown in Table 2.

$$E = \frac{30}{Q} \tag{19}$$

$$Q = W \times t \tag{20}$$

**Table 2.** Comparison of energy consumption of three catalytic processes.

| Catalyst | Process | Time Required to Remove 40% HPAM | Energy Consumption | Energy Efficiency |
|---|---|---|---|---|
| | | h | kW·h | mg HPAM/(kW·h) |
| CoTPyzPz | Photocatalysis | 4 | 1.32 | 22.73 |
| | Electrocatalysis | 6 | 1.38 | 21.74 |
| | Photoelectrocatalysis | 0.67 | 0.36 | 84.48 |
| CoTPyzPz/CB | Photocatalysis | 4 | 1.32 | 22.73 |
| | Electrocatalysis | 3.5 | 0.81 | 37.27 |
| | Photoelectrocatalysis | 0.51 | 0.27 | 110.99 |
| *t*Bu-CoTPyzPz | Photocatalysis | 2.1 | 0.69 | 43.29 |
| | Electrocatalysis | 6 | 1.38 | 21.74 |
| | Photoelectrocatalysis | 0.8 | 0.42 | 70.75 |
| *t*Bu-CoTPyzPz/CB | Photocatalysis | 2.1 | 0.69 | 43.29 |
| | Electrocatalysis | 1.5 | 0.35 | 86.96 |
| | Photoelectrocatalysis | 0.4 | 0.21 | 141.51 |

In the formulas, E represents energy efficiency, Q represents the energy consumption, W represents the total power of all electrical equipment for each catalytic process, and t represents the time taken for each process to remove 40% HPAM.

It can be clearly seen in Table 2 that the photocatalytic process had the highest energy consumption and the lowest energy efficiency when removing the same mass of HPAM; on the other hand, although the photoelectrocatalytic process had higher energy consumption per unit time than the other two, the treatment time was greatly shortened, so the energy efficiency was far better than the other two. Among them, the catalyst tBu-CoTPyzPz/CB had the highest energy efficiency. Under the experimental conditions, the energy efficiency of HPAM wastewater treatment reached 141.51 mg/(kW·h), which benefited from the increase in the total amount of active substances under the photoelectrocatalysis. The energy consumption comparison can be used as a basis for actual industrial production in the future.

To compare the three catalytic processes more intuitively, we list some related research results from recent years in Table 3. The degradation efficiency of the photoelectrocatalytic process is often better than that of the other two processes.

**Table 3.** Comparison of three catalytic processes recorded in the literature.

| Catalyst | Process | Pollutant | Degradation Rate | Reference |
|---|---|---|---|---|
| Imidazole Phthalocyanine | Photocatalysis | 2,3,4,5-Tetrachlorophenol | 80–90% | [5] |
| Zinc Phthalocyanine—Nanoporous Gold | Photocatalysis | 1,3-diphenylisobenzofuran | 92% | [33] |
| Metal phthalocyanines-$TiO_2$ nanoparticles | Photocatalysis | Ibuprofen | 90% | [34] |
| Flotation tailings particle electrode | Electrocatalysis | tetracycline | 75% | [35] |
| cerium doped Ti/nano-$TiO_2$/$PbO_2$ | Electrocatalysis | COD | 96.6% | [36] |
| Ti/$TiO_2$ nanotube | Photoelectrocatalysis | 5-fluorouracil | 100% | [37] |
| Nanopararticulate titania films/FTO | Photoelectrocatalysis | azo dye Basic Blue 41 | 95% | [8] |
| *t*Bu-CoTPyzPz/CB | Photoelectrocatalysis | HPAM | 94.55% | This paper |

## 3. Materials and Methods

### 3.1. Materials

CB (Vulcan XC-72R, Cabot Corporation, Boston, MA, USA), with a pore size of 30 nm and surface area of 232 $m^2$/g, was cleaned by ultrasound wave in deionized water for 30 min at room temperature, then dried at 75 °C in the vacuum drying oven. All the other chemicals, solvents and reagents were of AR grade, used as received, and purchased from the Sinopharm Chemical Reagent Company, Shanghai, China. The average molecular weight of HPAM was about $3 \times 10^6$ (the degree of hydrolysis of HPAM was about 16~18%).

### 3.2. Procedure for the Preparation of tBu-TPyzPzCo/CB

The preparation procedure of *t*Bu-TPyzPzCo/CB refers to the author's previous work [38]. In a typical experiment, *t*Bu-TPyzPzCo was synthesized by mixing pivalil (1) with 2,3-diaminomaleonitrile (2), urea (3), and cobalt chloride hydrate (4) in the presence of heat, of which a molar ratio of 1:1:4:0.5 from 1:2:3:4 obtained the best result. Pivalil was synthesized from pivaloin (5) by oxidation with chromic acid solution (6), based on the general procedure of Melvin S. Newman and A. Arkell [39]. The catalyst was first dissolved in DMF and ultrasonically dispersed. Then, carbon black was added, and the

sample continued to undergo ultrasonic dispersion for 1 h. The liquid was filtered, and the residue was washed with DMF until the filtrate became colorless. The residue was washed with distilled water in triplicate and dried under a vacuum at 70 °C to produce *t*Bu-TPyzPzCo/CB.

### 3.3. Catalytic Experiments

The photoelectrocatalytic reaction was carried out using self-made equipment, as shown in the schematic diagram in Figure 7. In this device, an open rectangular sink with a size of 16 cm × 8 cm × 15 cm was prepared as the main reactor, with an effective volume of 1.92 L. The light source for photocatalytic reaction was a 300 W xenon lamp (PLS -SXE300, Perfect Light, Beijing, China) which was placed above the main reactor. The anode and cathode materials were titanium and 304 stainless steel, respectively. The size of the electrode plate was 12 cm × 4 cm, and the distance between them was 5 cm. This study used a DC power supply (MN605D, Zhaoxin, China) with a voltage range of 0 to 60 V and a current range of 0 to 5 A. To maintain the concentration and temperature of the reaction solution during the reaction, we placed the reactor on a magnetic stirrer and used a peristaltic pump at the same time.

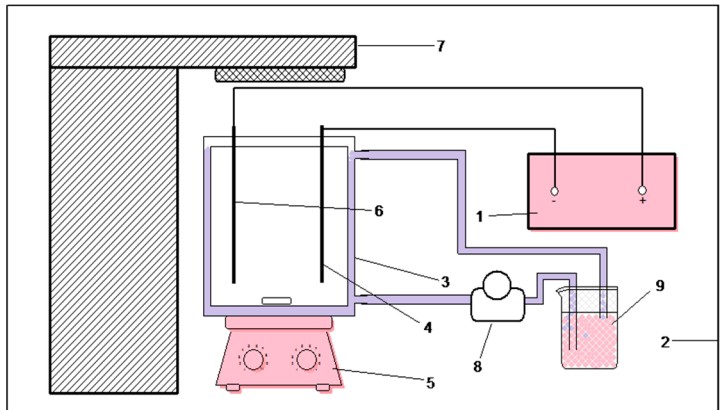

**Figure 7.** Photoelectrocatalytic experimental device diagram. 1 DC; 2 darkroom; 3 main reactor; 4 stainless steel cathode; 5 magnetic stirrer; 6 titanium anode; 7 xenon lamp; 8 peristaltic pump; 9 circulating cooling water.

In this paper, we studied the electrocatalysis, photocatalysis and photoelectrocatalysis processes of HPAM degradation under the same conditions. In each experiment, a certain concentration of HPAM solution was set as the reaction solution, and sodium sulfate ($Na_2SO_4$, 0.1 mol/L) was added as an electrolyte. The reaction solution was magnetically stirred for more than half an hour in the dark to reach the absorption equilibrium. While keeping the other aexperimental conditions unchanged, we turned on the xenon lamp/DC power supply and turned off the DC power supply/xenon lamp while studying the photocatalytic/electrocatalytic performances, respectively. The xenon lamp and DC power supply were turned on at the same time when the photoelectrocatalytic performance was studied. The viscosity and concentration of the HPAM solution were sampled every 20 min and measured immediately. The concentration was tested by the starch-cadmium iodide method [16,38] using an ultraviolet-visible spectrophotometer, and the viscosity was measured by a digital viscometer (DV-II + Pro, Brookfield, IL, USA). All measurements were performed at room temperature.

### 4. Conclusions

In summary, a *t*Bu-TPyzPzCo/CB composite catalyst was prepared by the ultrasonic impregnation method, and UV-vis, FTIR, and SEM were used to characterize the samples. Following the degradation of HPAM over the catalyst in a photoelectrocatalytic process, a

significant decrease was observed in the concentration and viscosity of the HPAM solution, which illustrates the excellent photoelectrocatalytic activity of the as-prepared catalysts.

Based on the research results of the photoelectrocatalytic process, the two signal catalytic processes were combined simply; however, we could also get significantly synergistic enhancement effects in addition to the two processes' respective degradation effects. This photoelectric combined process has many advantages, including utilizing the simple combination and rapid treatment for refractory wastewater, which exhibits a certain practical value for the improvement of existing facilities. Finally, based on the low energy consumption, relatively mild reaction conditions and good stability, low-cost, non-toxic catalysts are required to make the photoelectrocatalytic synergistic process a highly promising water treatment technology.

**Author Contributions:** Conceptualization, methodology, D.W., X.J. and C.Z.; validation, formal analysis, investigation, resources, data curation, writing—original draft preparation, D.W. and H.L.; writing—review and editing, visualization, supervision, X.J., Y.Z. and C.Z.; project administration, funding acquisition, D.W. All authors have read and agreed to the published version of the manuscript.

**Funding:** This research was supported by the Qingdao Postdoctoral Applied Research Project.

**Data Availability Statement:** Not applicable.

**Acknowledgments:** We gratefully acknowledge the North China Sea Environmental Monitoring Center of State Oceanic Administration and China University of Petroleum (East China) for supporting this work.

**Conflicts of Interest:** The authors declare no conflict of interest.

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
