# Peer review of "Comparison of Three Catalytic Processes in Degradation of HPAM by tBu-TPyzPzCo"

_catalysts, doi:10.3390/catal11020181_

Round 1
Reviewer 1 Report
In this paper, the authors designed a split photoelectrocatalytic reactor, using tert-butyl sub- stituted cobalt octaazaphthalocyanine supported by conductive carbon black (tBu- TPyzPzs/CB) as the catalyst, to study the degradation effect of hydrolytic polyacrylamide (HPAM) in different catalytic processes. I agree that this topic is very relevant for materials and environmental fields. This work is well written, however, it is poorer in characterization data compared to other previous works reported in literature. More characterization is needed and few points need to be clarified before I recommend its publication. Specific comments
1 – In introduction, clearly build your research hypothesis (straightforward question that is answerable by yes or no). I am not sure this is clear in the manuscript.
2 – Please underscore the novelty of your work in the introduction part. In the introduction, explain the main differences between your work and the ones found in literature
3 – Any word was written about the preparation of tBu-TPyzPzCo. The authors should explain in detail the steps about the material preparation, instead mentioning elsewhere in literature.
4 – In page 7…’’In each experiment, a certain concentration of HPAM solution was set as the reaction solution, and sodium sulfate (Na2SO4) was added as an electrolyte.’’…mention the right HPAM concentration used. Which amount of sodium sulfate ?
5 – The material was not characterized at all. It is very difficult to understand the treatment efficiency since there is no knowledge about the electrode structure, composition and morphologies. More characterization is needed, this can certainly support some statements written in the manuscript.
6 –The SEM analysis should be done, presented and discussed. This analysis can provide useful information about the adsorbent and about the adsorption efficiency process.
7 – FTIR is really important to characterize the materials and discuss its adsorption properties. The funcional groups play an important role in both electrochemical properties and adsorption process. Therefore it is needed.
8 – The other consideration is about to make a comparison, in a table, highlighting the main findings in the literature with the findings of this manuscript. The idea is to compare the effciency of the catalytic processes with others found in the literature.
9 – Conclusion is not well-written. Please rewrite it and focus on the main concept of the study. After execute the characterization analysis asked by this reviewer, please underscore the main results in the abstract and conclusion sections.
10 – Please cite at least 5 recent papers (2019-2020) from Catalysts.
Reviewer 2 Report
This work deals with the preparation of a cobalt complex of tetra-2,3-(5,6-di-tert-butyl-pyrazino) porphyrazine (tBu-TPyzPzCo), impregnated onto carbon black (CB) to prepare a supported catalyst (tBu-TpyzPzCo/CB) and it was used por the degradation of partially hydrolyzed polyacrylamide (HPAM) by photocatalysis, electrocatalysis and photoelectro-catalysis. I recommend its publication in Catalysts after a few points are addressed:
- No characterization of the cobalt complex is included and its preparation is only given by a reference of a previous work, it would be better to include a brief description of its preparation and of its characterization.
- References section must be revised since there are some mistakes: there are references with the first name complete and other with only their abbreviations (ref. 3, 4), ref. 19 is incomplete, some references have the abbreviations of the name of the journal finished in point (ref. 23, for instance), others not (ref. 25), etc..
Round 2
Reviewer 1 Report
The Authors responded thoroughly to my comments and recommendations. Therefore I recommend publication of the manuscript in Nanomaterials in the current form.